# Calculating Canopy Stomatal Conductance from Eddy Covariance Measurements, in Light of the Energy Budget Closure Problem

Richard Wehr[1], Scott. R. Saleska[1]

[1]Ecology and Evolutionary Biology, University of Arizona, Tucson, 85721, U.S.A.

*Correspondence to*: Richard Wehr (rawehr@email.arizona.edu)

**Abstract.** Canopy stomatal conductance is commonly estimated from eddy covariance measurements of the latent heat flux ($LE$) by inverting the Penman-Monteith equation. That method ignores eddy covariance measurements of the sensible heat flux ($H$) and instead calculates $H$ implicitly as the residual of all other terms in the site energy budget. Here we show that canopy stomatal conductance is more accurately calculated from EC measurements of both $H$ and $LE$ using the flux-gradient
equations that define conductance and underlie the Penman-Monteith equation, especially when the site energy budget fails to close due to pervasive biases in the eddy fluxes and/or the available energy. The flux-gradient formulation dispenses with unnecessary assumptions, is conceptually simpler, and is as or more accurate in all plausible scenarios. The inverted Penman-Monteith equation, on the other hand, contributes substantial biases and erroneous spatial and temporal patterns to canopy stomatal conductance, skewing its relationships with drivers such as light and vapor pressure deficit.

## 1 Introduction

Leaf stomata are a key coupling between the terrestrial carbon and water cycles. They are a gateway for carbon dioxide and transpired water and often limit both at the ecosystem scale (Jarvis and McNaughton, 1986). Although the many stomata in a plant canopy experience a wide range of micro-environmental conditions and therefore exhibit a wide range of behaviors at any given moment in time, it has proven useful in many contexts to approximate the canopy as a single 'big leaf' with a single
stoma (Baldocchi et al., 1991; Wohlfahrt et al., 2009; Wehr et al., 2017). That stoma is characterized by the canopy stomatal conductance to water vapor ($g_{sV}$), which can be defined as the total canopy transpiration divided by the transpiration-weighted average water vapor gradient across the many real stomata. This canopy stomatal conductance is not a simple sum of the individual leaf-level conductances and does not vary with time or environment in quite the same way as they do (Baldocchi et al., 1991); it is impacted, for example, by changes in the distribution of light within the canopy.


When the aerodynamic conductance to water vapor outside the leaf ($g_{aV}$) is greater than $g_{sV}$, the latter exerts a strong influence on transpiration, from which it can be inferred. The standard method is to calculate $g_{sV}$ from eddy covariance (EC) measurements of the latent heat flux ($LE$) via the inverted Penman-Monteith (iPM) equation (Monteith, 1965; Grace et al., 1995) — but the EC method and the iPM equation make a strange pairing. The original (not inverted) Penman-Monteith

equation was designed to estimate transpiration from the available energy ($A$), the vapor pressure deficit, and the stomatal and aerodynamic conductances. It was derived from simple flux-gradient relationships for $LE$ and for the sensible heat flux ($H$) but was formulated in terms of $A$ and $LE$ rather than $H$ and $LE$. Thus the inverted PM equation estimates $g_{sV}$ from $A$ and $LE$ rather than from $H$ and $LE$. EC sites, in contrast, measure $H$ and $LE$ but rarely assess $A$ in its entirety. True $A$ is net radiation ($R_n$) minus heat flux to the deep soil ($G$), minus heat storage ($S$) in the shallow soil, canopy air, and biomass. In wetland

ecosystems, heat flux by groundwater discharge ($W$) can also be important (Reed et al., 2018). While net radiation measurements are ubiquitous at EC sites, ground heat flux measurements are less common (Stoy et al., 2013; Purdy et al., 2016) and heat storage and discharge measurements are rare (Lindroth et al., 2012; Reed et al., 2018). As such it is common practice to simply omit $S$ and $W$ and sometimes $G$ from $A$ in the iPM equation.

In general, neither $S$ nor $G$ is negligible. Insufficient measurement of $S$ in particular has been shown (Lindroth et al., 2010; Leuning et al., 2012) to be a major contributor to the infamous energy budget closure problem at EC sites, which is that the measured turbulent heat flux $H + LE$ is about 20% less than the measured available energy $R_n - G$ on average across the FLUXNET EC site network (Wilson et al., 2002; Foken, 2008; Franssen et al., 2010; Leuning et al., 2012; Stoy et al., 2013). The other major contributor, which also impacts the iPM equation, is systematic underestimation of $H + LE$ by the EC method,

probably due to its failure to capture sub-mesoscale transport (Foken, 2008; Stoy et al., 2013; Charuchittipan et al., 2014; Gatzsche et al., 2018; Mauder et al., 2020). Leuning et al. (2012) assessed the relative contributions of $S$ and $H + LE$ to the closure problem using the fact that $S$ largely averages out over 24 hours while $R_n$, $H$, and $LE$ do not; thus $S$ contributes to the hourly but not the daily energy budget (Lindroth et al., 2010; Leuning et al., 2012). Analyzing over 400 site-years of data, they found that the median slope of $H + LE$ versus $R_n - G$ was only 0.75 when plotting hourly averages but went up to 0.9 when

plotting daily averages. This result suggests that for the average FLUXNET site, 60% of the energy budget gap is attributable to $S$ and 40% to $H + LE$. Depending on the depth at which $G$ is measured (which is not standard), $G$ might also average down considerably over 24 hours and thereby share some of the 60% attributed to $S$. Conversely, the part of $G$ that does not average out over 24 hours might share some of the 40% attributed to $H + LE$, as might $R_n$ and $W$. Part of that 40% might also be due to mismatch between the view of the net radiometer and the flux footprint of the eddy covariance tower. But $S$ and $H + LE$ are

the most likely sources of large systematic bias across sites.

The iPM equation is further impacted by how the underestimation of $H + LE$ is partitioned between $H$ and $LE$. While some studies have reported that underestimation of $H + LE$ roughly preserves the Bowen ratio ($B = H/LE$), others have reported that the failure to capture sub-mesoscale transport causes EC to underestimate $H$ more than $LE$ (Mauder et al., 2020) — a situation

that would benefit the iPM equation. Charuchittipan et al. (2014) quantified the preferential underestimation of $H$ relative to $LE$ using a simple formula based on the buoyancy flux, and a study of tall vegetation suggested that the formula holds when $B$ is high ($B > 2$) but that $B$ is instead preserved when it is low or moderate ($B < 1.5$) (Gatzsche et al., 2018). In the latest review

of the issue, Mauder et al. (2020) concluded that recent evidence "tends towards a partitioning somewhere between a buoyancy-flux-based and a Bowen-ratio-preserving" one.


To deal with the energy budget closure problem, Wohlfahrt et al. (2009) considered various schemes for correcting the fluxes in the iPM equation, following earlier recommendations that EC fluxes be corrected to close the energy budget in a more general context (Twine et al., 2000). All but one of the schemes in Wohlfahrt et al. (2009) involve attributing the half-hourly budget gap entirely to $A$ or entirely to $H + LE$, neither of which is generally realistic according to the subsequent results of

Leuning et al. (2012), mentioned above. The remaining option from Wohlfahrt et al. (2009) increases $H$ and $LE$ to close the long-term (e.g. daily or monthly) budget gap while preserving the Bowen ratio ($B$), which is in line with Leuning et al (2012) in that it attributes the long-term gap to EC and the remaining gap to storage.

Here we use data simulations to show that regardless of whether the energy budget gap is due to $A$ or $H + LE$, and regardless

of how the EC bias is partitioned between the buoyancy-flux and Bowen-ratio limits, stomatal conductance is more accurately obtained by direct application of the two simple flux-gradient (FG) equations on which the iPM equation is based than by use of the iPM equation itself. By using simulations, we can know the "true" target values and hence the absolute biases in $g_{sV}$. We also use our simulations to test the effects of perfect and imperfect eddy flux corrections, and of bias in the aerodynamic conductance outside the leaf. Lastly, we leave the simulations behind and show how the discrepancy between the FG and iPM

formulations impacts the retrieval of $g_{sV}$ over time using real measurements from a conifer forest. We present the FG and iPM formulations in Section 2, describe our methods for comparing them in Section 3, and report our findings in Section 4.

**2 Theory**

By definition, conductance is the proportionality coefficient between a flux and its driving gradient. In the case of $g_{sV}$, the flux is transpiration and the gradient is the vapor pressure differential across the "big-leaf" stoma. It is therefore relatively

straightforward to calculate $g_{sV}$ from the flux-gradient (FG) equations for transpiration and sensible heat (Baldocchi et al., 1991), rearranged as follows (Wehr and Saleska, 2015):

$$r_{sV} = \frac{e_s(T_L) - e_a}{RT_a E} - r_{aV} \qquad (1)$$

$$T_L = \frac{H r_{aH}}{\rho_a c_p} + T_a \qquad (2)$$

where $r_{sV}$ (s m$^{-1}$) is the stomatal resistance to water vapor, $r_{aV}$ is the aerodynamic resistance to water vapor (s m$^{-1}$), $r_{aH}$ is the aerodynamic resistance to heat (s m$^{-1}$), $E$ is the flux of transpired water vapor (mol m$^{-2}$ s$^{-1}$), $H$ is the sensible heat flux (W m$^{-}$

[2]), $T_a$ is the air temperature (K), $T_L$ is the effective canopy-integrated leaf temperature (K), $\rho_a$ is the density of (wet) air (kg m[-3]), $c_p$ is the specific heat capacity of (wet) air (J kg[-1] K[-1]), $e_a$ is the vapor pressure in the air (Pa), $e_s(T_L)$ is the saturation vapor pressure inside the leaf as a function of $T_L$ (Pa), and $R$ is the molar gas constant (8.314472 J mol[-1] K[-1]). The equation for the saturation vapor pressure (Pa) as a function of temperature (K) is (World Meteorological Organization, 2008):

$$e_s(T) = 611.2e^{\left(\frac{17.62(T-273.15)}{243.12+(T-273.15)}\right)} \tag{3}$$

The aerodynamic resistances describe the path between the surface of the "big leaf" and whatever reference point in the air at which $T_a$, $e_a$, $\rho_a$, and $c_p$ are measured. If that reference point is the top of an eddy flux tower, then that path includes the leaf boundary layer (through which transport is quasi-diffusive) as well as the canopy airspace and some above-canopy air (through which transport is turbulent). The turbulent eddy resistance ($r_e$) may be calculated by various methods that do not agree particularly well with one another (e.g. see Baldocchi et al., 1991; Grace et al., 1995; Wehr and Saleska, 2015), but is typically small in "rough surface" ecosystems like forests during the daytime, when $r_{aH}$ and $r_{aV}$ tend to be dominated by the leaf boundary layer resistances $r_{bH}$ and $r_{bV}$. An empirical model such as the one given in the Appendix can be used to calculate $r_{bH}$ as a function of wind speed and other variables. Using that model in a temperate deciduous forest, $r_{bH}$ was found to vary only between 8 and 12 s m[-1] (Wehr and Saleska, 2015), and so we simply take it to be constant at 10 s m[-1] here. The corresponding resistance to water vapor transport can be calculated from $r_{bH}$ via (Hicks et al., 1987):

$$r_{bV} = \frac{1}{f}r_{bH}\left(\frac{Sc}{Pr}\right)^{\frac{2}{3}} \tag{4}$$

where $Sc$ is the Schmidt number for water vapor (0.67), $Pr$ is the Prandtl number for air (0.71), and $f$ is the fraction of the leaf surface area that contains stomata ($f = 0.5$ for hypostomatous leaves, which have stomata on only one side, and $f = 1$ for amphistomatous leaves, which have stomata on both sides). The aerodynamic resistances to sensible heat and water vapor are then $r_{aH} = r_{bH} + r_e$ and $r_{aV} = r_{bV} + r_e$.

Finally, the stomatal conductance to water vapor (mol m[-2] s[-1]) is obtained from $r_{sV}$ by (Grace et al., 1995):

$$g_{sV} = \left(\frac{P}{RT_L}\right)\frac{1}{r_{sV}} \tag{5}$$

where $P$ is the atmospheric pressure.

The above FG theory is also the basis of the Penman-Monteith equation for a leaf (Monteith, 1965) and its inverted form (Grace et al., 1995), which can be expressed as:

$$r_{sV} = \frac{s(R_n - G - S - W - LE_{tr} - LE_{ev})r_{aH} + \rho_a c_p(e_s(T_a) - e_a)}{\gamma LE_{tr}} - r_{aV} \tag{6}$$

where $LE_{tr}$ is the latent heat flux associated with transpiration (W m$^{-2}$), $LE_{ev}$ is the latent heat flux associated with evaporation that does not pass through the stomata (W m$^{-2}$), $e_s(T_a)$ is the saturation vapor pressure of the air as a function of $T_a$ (Pa) rather than $T_L$, $s$ is the slope of the $e_s$ curve at $T_a$ (Pa K$^{-1}$), and $\gamma$ is the psychrometric constant at $T_a$ (Pa K$^{-1}$). $R_n$, $G$, $S$, and $W$ also have units of W m$^{-2}$. Latent heat flux is water vapor flux ($E$) times the latent heat of vaporization of water (about $44.1 \times 10^3$ J mol$^{-1}$).

The inverted PM equation is usually expressed in a slightly simpler form by neglecting the distinctions (a) between transpiration and evaporation, and (b) between the leaf boundary layer resistances to heat and water vapor. We retain those distinctions here in order to highlight two important points:

1. Absent a means to accurately partition the measured eddy flux of water vapor into transpiration and non-stomatal evaporation (e.g. from soil or wet leaves), the FG and iPM equations are applicable only when evaporation is negligible, which is a difficult situation to verify but does occur at particular times in particular ecosystems (see, e.g., Wehr et al., 2017).

2. Setting $r_{bV} = r_{bH}$ instead of using Eq. (4) is a good approximation for amphistomatous leaves (stomata on both sides) but a poor approximation for the more common hypostomatous leaves (stomata on only one side) (Schymanski and Or, 2017). Indeed, we find that if $r_{bV}$ is set equal to $r_{bH}$ for hypostomatous leaves, the iPM equation underestimates $g_{sV}$ by about 10% (depending on the relative resistances of the stomata and boundary layer) even when the site energy budget is closed.

Note that the iPM equation can be derived from the FG equations by invoking energy balance to replace $H$ with $A - LE$ in Eq. (2) and then linearizing the Clausius-Clapeyron relation to eliminate leaf temperature:

$$s \approx \frac{e_s(T_a) - e_s(T_L)}{T_a - T_L} \Rightarrow e_s(T_L) \approx e_s(T_a) - s(T_a - T_L) = e_s(T_a) + s\left(\frac{(A - LE)r_{aH}}{\rho_a c_p}\right) \tag{7}$$

This psychrometric approximation has been shown to cause significant bias and incorrect limiting behavior in the Penman-Monteith equation (McColl, 2020). McColl (2020) derived a similar, alternative equation that remedies those problems but still uses measurements of $A$ instead of $H$. The psychrometric approximation and the substitution for $H$ are the only two

differences between the FG and iPM formulations. Both formulations rely on the same water flux measurements to estimate transpiration, both approximate the canopy as a 'big leaf', and both use the same estimate of aerodynamic resistance.

**3 Methods**

Our analysis consisted of two parts: simulations and real data analysis. The simulations were designed to unambiguously demonstrate the impact of flux measurement biases and the resultant energy budget gap on FG and iPM calculations of $g_{sV}$, as well as to test the sensitivity of $g_{sV}$ to bias in the estimated aerodynamic resistance outside the leaf; they are described in Section 3.1. The real data analysis was designed to assess the magnitude and temporal variation of the discrepancy between the FG and iPM formulations in a real forest and is described in Section 3.2.

**3.1 Simulations**

We assessed the proportional bias in $g_{sV}$ calculated via the iPM and FG formulations by simulating observations and using them to estimate $g_{sV}$. The simulations were of three snapshots in time roughly typical of midday in three different ecosystems: a temperate deciduous forest in July (the Harvard Forest in Massachusetts, USA; Wehr et al., 2017), a tropical rainforest in May (the Reserva Jaru in Rondônia, Brazil; Grace et al., 1995), and a tropical savannah in September (Virginia Park in
Queensland, Australia; Leuning et al., 2005). The purpose of including three different ecosystems was to test the FG and iPM formulations across a broad range of environmental and biological input variables (especially Bowen ratios), not to provide a lookup table of quantitative $g_{sV}$ corrections for other sites. The particular sites and time periods within each ecosystem were chosen merely for convenience, as the requisite variables were readily obtainable from the literature or from our past work.

The simulations began by setting the "true" target values of all the variables involved; in other words, their values without any simulated measurement error. To keep these values realistic, we started with approximate observed fluxes and conditions obtained from the papers cited above or from our own work at the Harvard Forest (Table 1), with the precise values of $H$ and $LE$ chosen to satisfy $B$ and energy balance. These fluxes and conditions were then used to calculated the true target $g_{sV}$ using the FG equations (Eqs. (1-5)). As the fluxes in Table 1 close the energy budget perfectly, the FG and iPM equations are
interchangeable for this step of the simulations apart from the psychrometric approximation (Eq. (7)), which causes a small but significant (~5%) positive bias in iPM-derived $g_{sV}$. That bias is the reason why iPM-derived $g_{sV}$ does not quite converge on the true value even when the entire energy budget gap is due to the EC fluxes and those fluxes are perfectly corrected (see Fig. 2). Thus we could have instead used the iPM equation (Eq. (6)) to set the true $g_{sV}$ and obtained similar results, except that the psychrometric approximation bias would have appeared, incorrectly, to afflict the FG results instead of the iPM results.

Next, we simulated a wide range of measurement bias scenarios, each with a 20% gap in the energy budget (the FLUXNET average). The simulations were explored along three main axes of variation:

Variation 1. The energy budget gap was variously apportioned between measurement bias in $A$ and measurement bias in $H + LE$. The measurement bias in $H + LE$ was applied proportionally to $H$ and $LE$ so as to preserve the true Bowen ratio. All other variables were unbiased.

Variation 2. Measurements of $H$ and $LE$ biased the Bowen ratio by varying amounts while the apportioning of the energy budget gap between $A$ and $H + LE$ was fixed at the FLUXNET average (60% $A$, 40% $H + LE$). All other variables were unbiased.

Variation 3. Estimates of the aerodynamic conductance outside the leaf were biased by varying amounts while the apportioning of the energy budget gap between $A$ and $H + LE$ was fixed at the FLUXNET average and the measurements of $H$ and $LE$ preserved the true Bowen ratio. All other variables were unbiased.

For each measurement bias scenario, we used the FG and iPM formulations to calculate $g_{sV}$ from the simulated (usually erroneous) eddy flux measurements, from perfectly corrected eddy flux measurements, and from eddy flux measurements adjusted to close the long-term energy budget while preserving the Bowen ratio, as proposed in Wohlfahrt et al. (2009). In our simulations of a single point in time, the latter adjustment was represented by increasing $H$ and $LE$ proportionally such that $H + LE$ became equal to the true value of $A$. Such an adjustment restores the true eddy fluxes if their measurements did not bias the Bowen ratio, and was therefore redundant with the perfect correction for Variation-1 and Variation-3. Conversely, the perfect correction was of no interest for Variation-2, as it removes all bias in the Bowen ratio.

**3.2 Analysis of Real Measured Time Series**

In order to show how the FG and iPM methods differ in a real forest over the diurnal cycle, we calculated time series of $g_{sV}$ from real hourly measurements at Howland Forest recorded in the AmeriFlux EC site database (Site US-Ho1; Hollinger, 1996). Howland Forest is a mostly coniferous forest in Maine, USA (45°12'N, 68°44'W), which we chose for its intermediate Bowen ratio, for variety, and otherwise for convenience. In addition to using the original measured fluxes, we also calculated $g_{sV}$ after adjusting the eddy fluxes using the long-term energy budget closure scheme proposed by Wohlfahrt et al. (2009) – the same flux adjustment scheme we tested in our simulations. For this scheme at Howland Forest, we computed the slope of $H + LE$ versus $R_n – G$ from a plot of all 24-hour averages in the summer of 2014 and then divided both $H$ and $LE$ by that slope.

To minimize the influence of non-stomatal evaporation, we focused on two sunny midsummer days more than 24 hours after the last rain (July 25-26, 2014). Because our aim was to show the relative bias between the FG and iPM methods rather than to obtain the most accurate possible estimate of $g_{sV}$, we used the constant and roughly appropriate values $r_{bH} = 10$ s m$^{-1}$ and $r_e = 0$ as in our simulations, rather than calculating values for these aerodynamic resistances from the data according to models like that in the Appendix. Given that conifer forests are very rough surfaces and that the two daylight periods under consideration were windy with strong turbulent mixing (wind speed > 3 m s$^{-1}$ and friction velocity > 0.6 m s$^{-1}$ from late morning

through late afternoon), it is almost certain that the aerodynamic resistance was much less than the stomatal resistance and therefore that the FG and iPM equations were insensitive to $r_{bH}$ and $r_e$ (see Section 4.1).

## 4 Results and Discussion

### 4.1 Absolute Biases Revealed by Simulations

Our simulations indicate that the flux-gradient formulation is substantially more accurate than the inverted Penman-Monteith equation regardless of the cause and magnitude of the energy budget gap, and regardless of the ecosystem type.

Figure 1 shows bias in $g_{sV}$ versus the relative contribution of eddy flux bias to the hourly energy budget gap (the remainder of the gap being due to bias in the available energy). This figure follows Variation-1 from Section 3.1, which assumes that eddy
flux measurements preserve the true Bowen ratio. Here the FG formulation (solid black lines) is always more accurate than the iPM formulation (solid red lines) because regardless of whether the gap is due to negative measurement bias in $A$ or in $H$ + $LE$, the iPM equation implicitly overestimates $H$ (as the residual of the other fluxes) and therefore the leaf temperature and therefore the water vapor gradient, which exacerbates underestimation of the conductance. In other words, it is better to have both $LE$ and $H$ underestimated (as in the FG equations) than to have $LE$ underestimated and $H$ overestimated (as in the iPM
equation). The dashed lines in Fig. 1 show results calculated using eddy fluxes that have been corrected back to the true values (as studies have aimed to do), in which case the FG formulation becomes unbiased while the iPM equation still suffers from bias in $A$ and from the psychrometric approximation. Figure 2 clarifies the contributions of $LE$, $A$, and the psychrometric approximation (Eq. (7)) to bias in the iPM equation.

Comparison of Figs. 1a, 1b, and 1c reveals that the qualitative relationships in Fig. 1 do not depend on the values of the environmental and biological variables in Table 1, but the severity of the bias in $g_{sV}$ does. The bias in $g_{sV}$ is also proportional to the relative energy budget gap, i.e. $(H + LE)/(R_n - G)$, and will therefore be larger (smaller) than shown here at sites with gaps larger (smaller) than 20%. Because the bias in $g_{sV}$ varies with environmental and biological site characteristics, it will lead to erroneous spatial patterns in $g_{sV}$ and to erroneous relationships with potential drivers.

As noted in the introduction, pervasive eddy flux biases likely preserve the true Bowen ratio in some but not all circumstances. Thus Figure 3 shows bias in $g_{sV}$ versus bias in the measured Bowen ratio. This figure follows Variation-2 from Section 3.1, which assumes that 40% of the energy budget gap is due to the eddy fluxes (which is the FLUXNET average). The FG formulation (solid black lines) remains more accurate than the iPM equation (solid red lines) everywhere between the
250 buoyancy-flux-based and Bowen-ratio-preserving limits, except very close to the buoyancy-flux-based limit in the high-$B$ tropical savanna. The iPM equation becomes nearly unbiased in that situation because its inherent assumption that the energy budget gap is due entirely to $H$ becomes nearly true; moreover, the small remaining bias due to underestimation of $LE$ is offset

by bias from the psychrometric approximation, which has opposite sign (see Fig. 2). The dotted lines in Fig. 3 show results calculated using eddy fluxes that have been adjusted to close the long-term energy budget while preserving the (erroneously measured) Bowen ratio (if the eddy fluxes were perfectly corrected as in Fig. 1, there would be no variation along the abscissa for any method in this figure). We include this mis-correction because it is the most likely adjustment to be applied to eddy fluxes in practice, whether it is appropriate or not. It favors the FG formulation when the Bowen ratio bias is small, and begins to favor the iPM equation as that bias increases — and it illustrates how improper correction of the eddy fluxes can make the bias in $g_{sV}$ worse.

Aside from the energy budget gap, another potentially important source of bias in the FG and iPM equations is the aerodynamic resistance ($r_{aH} = r_{bH} + r_e$ and $r_{aV} = r_{bV} + r_e$). Estimates of the aerodynamic resistance come from models of the leaf boundary layer (such as that in the Appendix) and of micrometeorology (see Baldocchi et al., 1991). These models are based on established theory and careful experiments but involve many parameters and assumptions that are not well constrained in real ecosystems. As a result, the uncertainty in the aerodynamic resistance is generally unknown. Figure 4 shows how bias in $g_{sV}$ is impacted by a range of plausible biases in the estimated boundary layer resistance (a factor of 2 in either direction), following Variation-3 from Section 3.1. Here the apportioning of the energy budget gap between $A$ and $H + LE$ is fixed at the FLUXNET average and the measurements of $H$ and $LE$ preserve the true Bowen ratio. Especially when the Bowen ratio is far from 1 (Fig. 4b, c), plausible bias in the boundary layer estimate can lead to large biases in $g_{sV}$ regardless of whether the FG or iPM formulation is used. On the other hand, when $B = 0.6$ in the temperate forest (Fig. 4a), the effects of the boundary layer on sensible and latent heat roughly cancel one another out in the FG formulation, so that $g_{sV}$ is insensitive to the boundary layer estimate. Biases in the boundary layer resistance rarely make the iPM equation more accurate than the FG equations.

If the aerodynamic resistance outweighs the stomatal resistance, then transpiration is insensitive to the stomata and it is inadvisable to try to retrieve $g_{sV}$ from measurements of the water vapor flux. Essentially, transpiration does not carry much information about the stomata in this case, and so the uncertainty in retrieved $g_{sV}$ would be large regardless of whether the FG or iPM formulation was used. This is the 'decoupled' limit described by Jarvis and McNaughton (1986) and the 'calm limit' described by McColl (2020). Comparison of Figure 5 to Fig. 4a demonstrates that as the ecosystem moves toward this limit, the sensitivity of $g_{sV}$ to bias in the aerodynamic resistance increases as expected; however, if the aerodynamic resistance is estimated perfectly ($r_{bH}$ bias = 0, marked by the vertical grey line), then the FG equations actually become slightly more accurate in this limit while the iPM equation becomes substantially more biased. The reason is that a large aerodynamic resistance impedes the exchange of heat and so increases the leaf temperature, which increases the saturation vapor pressure inside the leaf by an even greater factor (according to the nonlinear Clausius-Clapeyron relation). Thus transpiration actually increases and the Bowen ratio approaches zero, so that underestimation of $H$ becomes unimportant but underestimation of $LE$ becomes more important. The psychrometric approximation also becomes poorer in this situation because it is a linearization of the Clausius-Clapeyron relation (McColl, 2020).

## 4.2 Relative Biases over Time in a Real Forest

Figure 6 compares the diurnal patterns of $g_{sV}$ calculated from real measurements at Howland Forest (Hollinger, 1996) using the FG (black) and iPM (red) formulations. Solid lines show results based on the original EC fluxes and dotted lines show
results based on adjusted EC fluxes that closed the long-term energy budget while preserving the measured Bowen ratio (the same adjustment as shown in Fig. 3). As usual at EC sites, heat storage was not measured and was therefore omitted from the iPM equation. If the bias in $A$ did indeed average out at the monthly timescale, and if the measured Bowen ratio and estimated aerodynamic resistances were accurate, then the true values of $g_{sV}$ in Fig. 6 should be those obtained using the FG formulation with adjusted EC fluxes (dotted black lines). That flux adjustment was relatively small at this site in the summer of 2014: the
slope of hourly $H + LE$ versus hourly $R_n - G$ was only 0.63, while the slope using daily data was 0.92, suggesting that 78% of the hourly energy budget gap was due to the omission of $S$ and only 22% was due to EC. Besides the expected negative bias in the iPM approach, Fig. 6 shows that the iPM and FG formulations claim noticeably different diurnal patterns for $g_{sV}$. In particular, the iPM equation gives substantially lower values than the FG formulation through the morning and early afternoon but then converges on the FG formulation in the late afternoon. The diurnal curve obtained from the iPM equation is therefore
too flat, leading to an understated picture of the response of $g_{sV}$ to the vapor pressure deficit (which peaks in the afternoon), and/or to an exaggerated picture of the saturation of $g_{sV}$ at high light. This time-varying discrepancy between the FG and iPM approaches can be explained by the fact that $S$ (and therefore negative bias in the iPM equation) generally peaks in the late morning and approaches zero in the late afternoon (Grace et al., 1995; Lindroth et al., 2010), as reflected in the energy budget gap shown in the bottom panel of Fig. 6 (grey shading).

## 5 Conclusion

We have shown that for the purpose of determining canopy stomatal conductance at eddy covariance sites, the inverted Penman-Monteith equation is an inaccurate and unnecessary approximation to the flux-gradient equations for sensible heat and water vapor. Incomplete measurement of the energy budget at EC sites causes substantial bias and misleading spatial and temporal patterns in canopy stomatal conductance derived via the iPM equation, even after attempted eddy flux corrections.
The biases in iPM stomatal conductance vary between 0 and ~30% depending on the time of day and the site characteristics, resulting in erroneous relationships between stomatal conductance and driving variables such as light and vapor pressure deficit. Models trained on those relationships can be expected to misrepresent canopy carbon-water dynamics and to make incorrect predictions.

In theory, the FG equations are mathematically equivalent to the iPM equation aside from the relatively minor psychrometric approximation in the latter. In practice, however, errors in $H$ and $LE$ push $g_{sV}$ in opposite directions and so it is crucial that the FG equations receive underestimates of $H$ and $LE$ whereas the iPM equation implicitly overestimates $H = A - LE$ from overestimates of $A$ ($= R_n - G - S - W$) and underestimates of $LE$. As a result, bias in $g_{sV}$ tends to be only about half as large in

the FG equations as in the iPM equation. Moreover, if the eddy fluxes can be properly corrected, then the FG equations become
unbiased while the iPM equation still suffers from bias in $A$.

Unfortunately, there does not appear to be a universally appropriate method for correcting the eddy fluxes at present. When the Bowen ratio is low or moderate in tall vegetation like forests, the published evidence supports increasing $H$ and $LE$ proportionally to close the long-term energy budget. However, when the Bowen ratio is high, the evidence suggests that $H$
needs a disproportionally larger correction than $LE$. In that case, we have shown that a Bowen-ratio-preserving correction can make the bias in $g_{sV}$ worse.

Our results suggest that future studies should use the FG equations in place of the iPM equation, and that published results based on the iPM equation may need to be revisited. It also motivates further work to determine a general and reliable
framework for correcting the measured fluxes of sensible and latent heat at eddy covariance sites.

**Appendix: An empirical formula for the leaf boundary layer resistance to heat transfer**

The canopy flux-weighted leaf boundary layer resistance to heat transfer from all sides of a leaf or needle (s m$^{-1}$) can be estimated approximately as (McNaughton and Hurk, 1995; Wehr et al., 2015):

$$r_{bH} = \frac{150}{\text{LAI}} \sqrt{\frac{L}{u_h}} \int_0^1 e^{\alpha(1-\zeta)/2} \phi(\zeta) d\zeta \qquad \text{(A1)}$$

where LAI is the single-sided leaf area index, $L$ is the characteristic leaf (or needle cluster) dimension (e.g. 0.1 m), $u_h$ is the mean wind speed (m s$^{-1}$) at the canopy top height $h$ (m), $\zeta$ is height normalized by $h$, $\phi(\zeta)$ is the vertical profile of the heat source (which can be approximated by the vertical profile of light absorption) normalized such that $\int_0^1 \phi(\zeta) d\zeta = 1$, and $\alpha$ is
the extinction coefficient for the assumed exponential wind profile:

$$\frac{u(\zeta)}{u_h} = e^{\alpha(\zeta-1)} \qquad \text{(A2)}$$

where $\alpha = 4.39 - 3.97e^{-0.258\text{LAI}}$. The wind speed at the top of the canopy can be obtained from Eq. (A2) with $\zeta$ set to
correspond to the wind measurement height atop the flux tower.

**Code availability**

The R code used for the simulations and the Igor Pro code used for the Howland Forest data analysis are freely available in the Dryad data archive under the digital object identifier doi:10.5061/dryad.h44j0zpgp (Wehr and Saleska, 2020).

**Author contribution**

RW conceived and designed the study, wrote the software code, performed the simulations, and prepared the manuscript with contributions from SRS.

**Competing interests**

The authors declare that they have no conflict of interest.

**Acknowledgements**

This work was supported by the National Science Foundation (award #1754803). Funding for AmeriFlux data resources was provided by the U.S. Department of Energy's Office of Science. The Howland Forest data was produced under the supervision of Dr. David Hollinger.

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

| Variable | Temperate Forest (representing July at the Harvard Forest, U.S.A., 42°32'N, 72°10'W) | Tropical Forest (representing May at Reserva Jaru, Brazil, 10°5'S, 61°57'W) | Tropical Savannah (representing September at Virginia Park, Australia, 35°39'S, 148°9'E) |
|---|---|---|---|
| Bowen Ratio, $B$ | 0.6 | 0.35 | 8 |
| Sensible Heat Flux, $H$ (W m$^{-2}$) | 236 | 140 | 418 |
| Latent Heat Flux, $LE$ (W m$^{-2}$) | 394 | 400 | 52 |
| Net Radiation, $R_n$ (W m$^{-2}$) | 700 | 600 | 600 |
| Heat Storage, $S + G$ (W m$^{-2}$) | 70 | 60 | 130 |
| Air Temperature, $T_a$ (K) | 298 | 296 | 303 |
| Atmospheric Vapor, $e_a$ (Pa) | 1700 | 1800 | 1800 |

For all sites, $W = 0$ W m$^{-2}$, $r_{bH} = 10$ s m$^{-1}$, $r_e = 0$ s m$^{-1}$, $P = 101325$ Pa, $f = 0.5$.

**Table 1. Values of environmental and biological variables used in the error simulations (representing midday).**

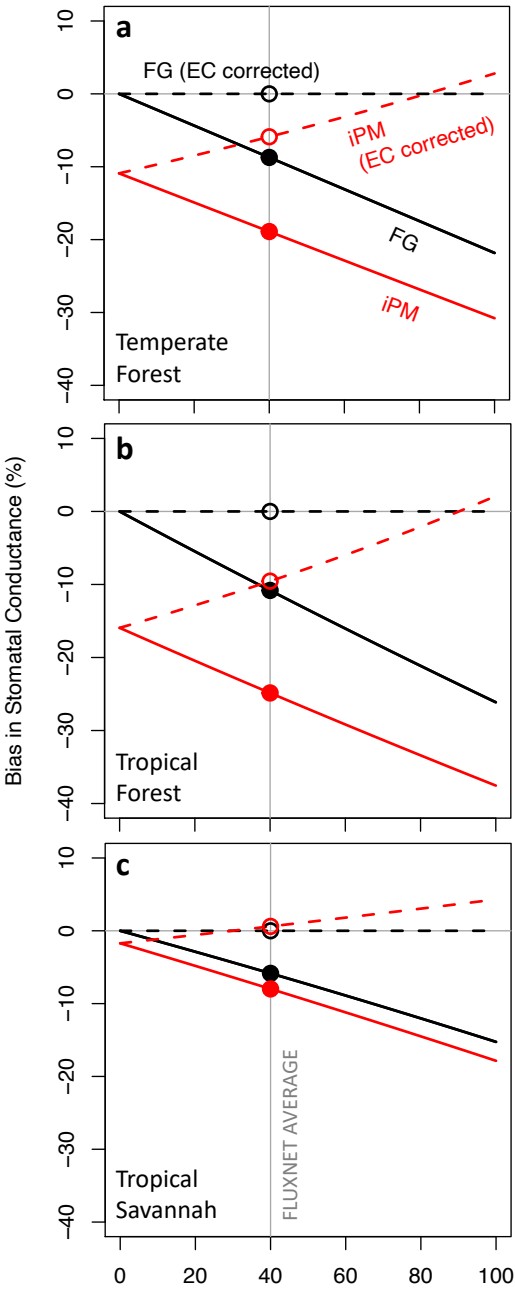

**Figure 1. Proportional bias in canopy stomatal conductance obtained from the flux-gradient (FG, black) and inverted Penman-Monteith (iPM, red) formulations versus the fraction of the hourly energy budget gap caused by bias in the eddy fluxes rather than by bias in the available energy. Solid lines show results without eddy flux correction and dashed lines show results with perfectly corrected eddy fluxes. The average estimated contribution of eddy flux bias to the budget gap across FLUXNET is indicated by the grey vertical line (Leuning et al., 2012). Circles highlight where the various lines cross the FLUXNET average.**


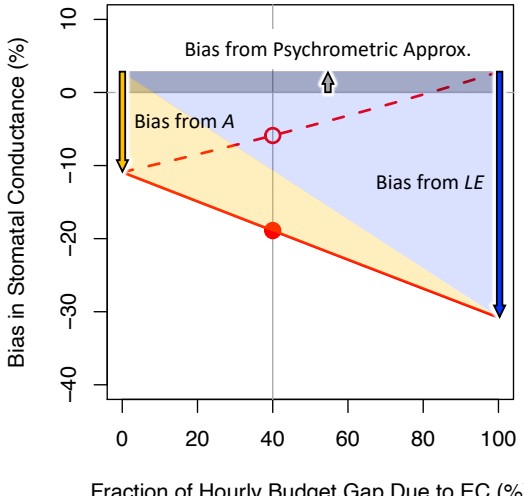


**Figure 2. Inverted Penman-Monteith results from Fig. 1a, annotated to indicate the various sources of bias.**

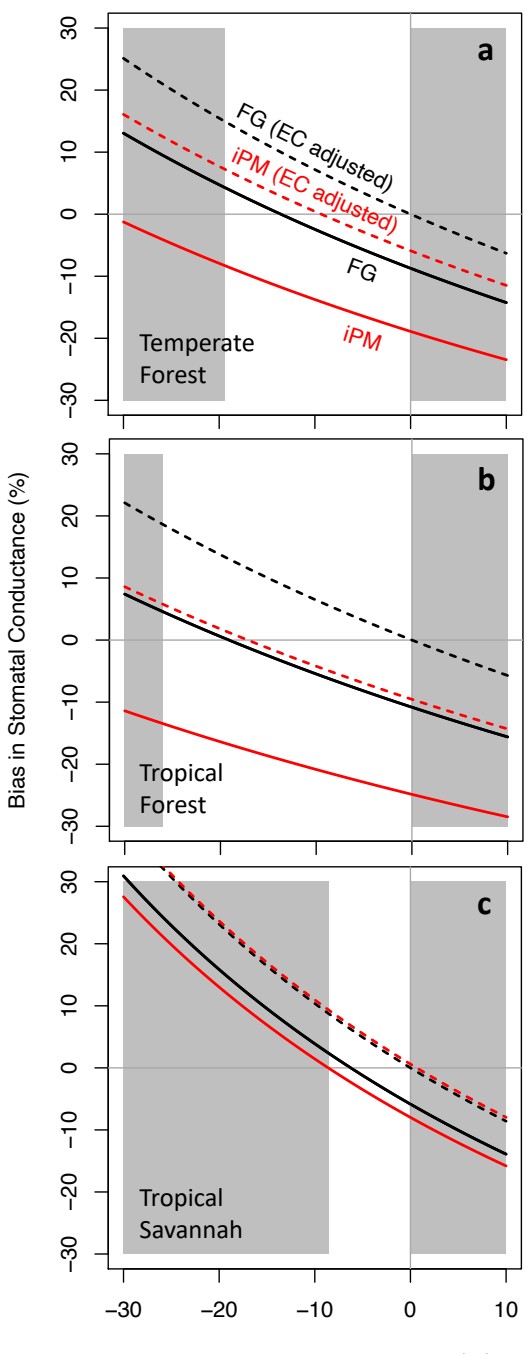

Figure 3. Proportional bias in canopy stomatal conductance obtained from the flux-gradient (FG, black) and inverted Penman-Monteith (iPM, red) formulations versus proportional bias in the measured Bowen ratio. Solid lines show results without eddy flux correction, and dotted lines show results with the eddy fluxes adjusted to close the long-term energy budget while preserving the (erroneously measured) Bowen ratio. The unshaded region denotes the plausible range of pervasive bias, which is bounded by the buoyancy-flux-based and Bowen-ratio-preserving limits (see text).


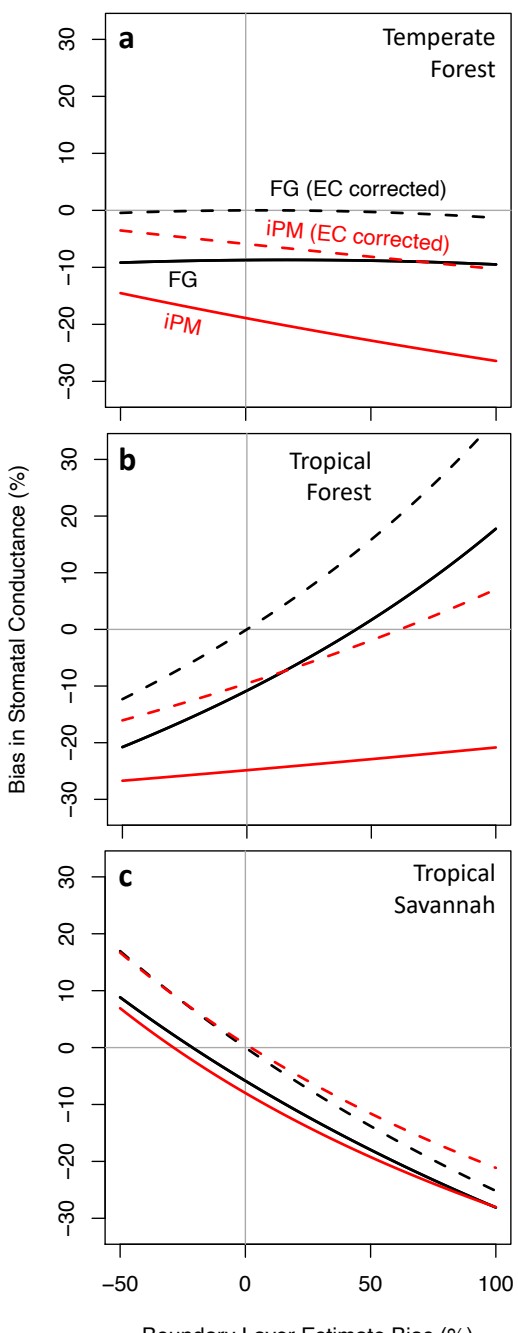


**Figure 4. Proportional bias in canopy stomatal conductance obtained from the flux-gradient (FG, black) and inverted Penman-Monteith (iPM, red) formulations versus proportional bias in the estimated boundary layer resistance. Solid lines show results without eddy flux correction and dashed lines show results with perfectly corrected eddy fluxes.**

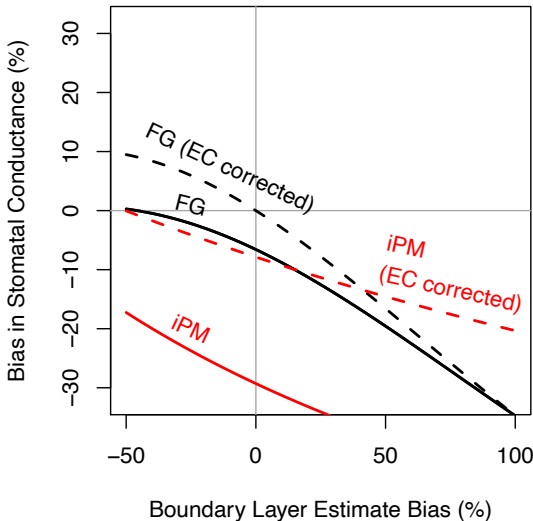

**Figure 5. Same as Fig. 4a, but with true boundary layer resistance increased to make the aerodynamic and stomatal conductances to water vapor equal, simulating very calm atmospheric conditions and increasing the sensitivity of the FG and iPM equations to the value used for the boundary layer resistance.**


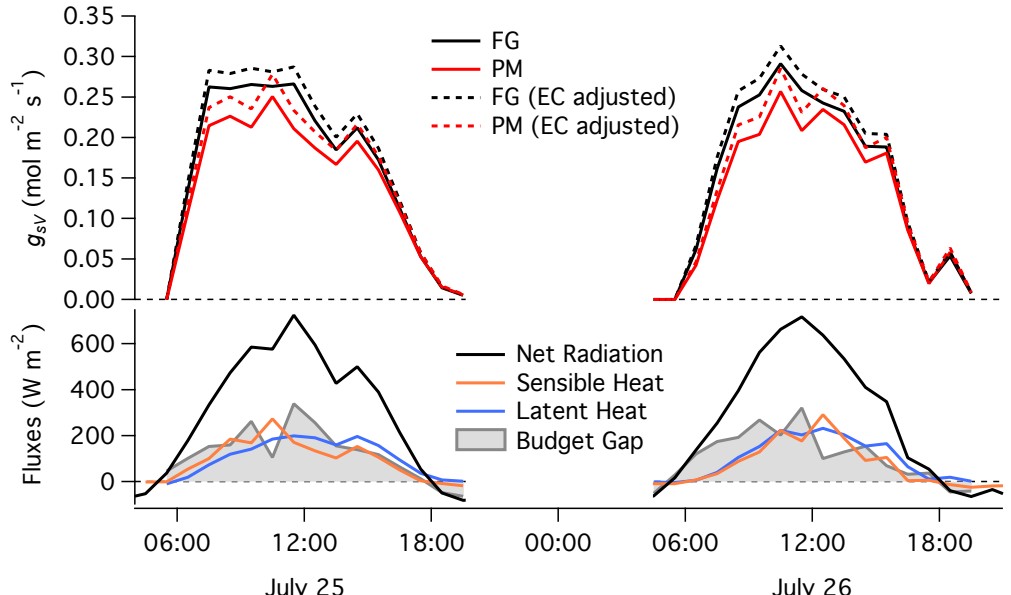

**Figure 6. Top panel: hourly canopy stomatal conductance to water vapor calculated at Howland Forest (Hollinger, 1996) over two days in 2014 by the same approaches as in Fig. 3. Bottom panel: measured energy fluxes and budget gap.**