# Peer review of "Calculating Canopy Stomatal Conductance from Eddy Covariance Measurements, in Light of the Energy Budget Closure Problem"

_Biogeosciences, 2020_

## Referee Comment (RC1) · Anonymous Referee #1 · 12 Jun 2020

The manuscript by Wehr and Saleska revisited the calculation of canopy stomatal conductance from eddy covariance measurements using the Penman-Monteith equation. They specifically focused on how the energy imbalance issue and the different corrections of this issue could impact the calculation of canopy stomatal conductance. They proposed a new approach that combines the flux-gradient formulation and correction of the energy imbalance while preserving the Bowen ratio. Overall, this is an innovative study and should be considered for publication in the Biogeosciences after some revision. I have a few general comments and suggestions. [1] The readability of this paper

could be improved. I found that Methods, Results, and Discussion are difficult to follow. I had to go back and forth several times to find the necessary details and information. I understand this type of paper might not necessarily follow the same structure of a typical research paper, but I suggest the authors should clearly ad structurally lay out the data used and steps taken upfront. Potentially, an overview paragraph summarizing the study design, a table listing the different simulation scenarios, and/or a more explicit subtitle might also help readers. Figure legends, especially Figure 1 & 2, should be more self-explanatory. [2] Several recent studies suggested that the energy imbalance issue was likely caused by mesoscale or secondary circulations instead of instrumental or other local sources, and H and LE might be influenced disproportionally (Mauder et al., 2020; Xu et al., 2020). It might still be an open question, but I suggest the authors taking that into consideration. [3] It's a bit puzzling to me about one of the key arguments – the preferred use of flux gradient equations instead of Penman-Monteith equation. I think ultimately the main difference resulted from how the energy imbalance was treated and/or how the total available energy was partitioned. The psychrometric approximation should have only marginal influence, right? Or, do the authors imply anything additionally? For example, some studies used available energy (LE+H) or adjusted total energy in the Penman-Monteith equation. Would it be sufficient enough? Mauder, M., Foken, T. and Cuxart, J., 2020. Surface-Energy-Balance Closure over Land: A Review. Bound-Lay Meteorol. Xu, K., Sühring, M., Metzger, S., Durden, D. and Desai, A.R., 2020. Can Data Mining Help Eddy Covariance See the Landscape? A Large-Eddy Simulation Study. Bound-Lay Meteorol, 176(1): 85-103.

---

## Referee Comment (RC2) · Anonymous Referee #2 · 23 Aug 2020

Wehr and Saleska identify the implication of residuals in the energy balance at flux sites and its implications on the calculations of stomatal conductance. The authors are well posed to tackle this problem given their previous work on stomatal conductance modelling. While the issue is important, they falter in the motivation of the study and ignore very carefully laid out theory regarding movement of water between leaves and the atmosphere. The authors state that the Penman- Monteith equation was developed because sensible heat fluxes (H) were immeasurable at the time (while those of latent heat were not). This is not true, since both H and LE have previously been inferred

from flux-gradient approaches or Bowen- ratio based approaches, and were prone to similar errors. In the abstract the authors note "...even though H is measured at least as accurately as LE at every EC site while the rest of the energy budget almost never is". This is also needs to be rephrased. Components of net radiation (Rnet) are routinely measured, and in fact, this is generally a more reliable measurement than the eddy flux of latent and sensible heat, which is prone to well-known errors (e.g. poor turbulence). However, putting the issue of instrumentation aside, the differences between H + LE + G and Rnet is often due to differences in "fetch" or missing large eddies in tall forest canopies. 4 channel net radiometers are placed adjacent to the towers, and can be influenced by the tower itself, whereas measurements from IRGAs are highly contingent on the footprint. The authors note correctly that G (soil heat fluxes) are not universally measured, and areas sampled may not be representative of the average soil heat flux from the site.

Coming to the main argument of the paper, the Penman-Monteith equation accounts for the fact that water loss from landscapes is controlled by biotic and abiotic factors. Leaves can reduce the width of stomatal apertures to limit water loss. However, the total evaporation from terrestrial ecosystems, especially leaves, is also a function of available energy (Rnet). This framework was developed in a seminal paper by Paul Jarvis and Keith McNaughton (1986) where they proposed a decoupling coefficient that determines the extent to which transpiration is "stomatally imposed". By using a simple flux-gradient theory the authors imply that transpiration is totally stomatally imposed. This argument is likely to work in tall rough canopies where exchange of momentum (and therefore scalars) with the well-mixed air above is efficient. However, at sites where roughness is low and canopies are homogenous, this approach is more likely to lead to erroneous estimates of stomatal conductance, since it is the available energy that will dominate the amount of water vapour that is lost to the atmosphere. Of course, to an extent, this problem is mitigated by also inferring boundary layer conductance and sharing the limitation of water loss between stomata and resistance imposed by the boundary layer.

Penman-Monteith never intended to solve for stomatal conductance, rather to estimate water loss from vegetated canopies in a way that eliminated the need to know surface variables e.g. surface temperature. The method includes the parametrization of a "surface conductance" which really is somewhat of an emergent property since its accounts for a cumulative effect of "all stomata" but also canopy structure and coupling (much like a canopy scale stomatal conductance or gsv in the current manuscript).

I do really appreciate the careful analyses that the authors have performed and a nice example of this is highlighting the errors in the psychometric approximation, when FG and PM methods are considered equivalent.

Thus, in my opinion, the authors should need to reconcile errors due to flux-gradient approaches with Jarvis and McNoughton (1986). I would then consider this a very significant contribution to the literature and fit to be published in Biogeosciences.

A minor note on figures: I think Figures 1 and 2 are a little complicated and could be simplified. It might help to even show figure 3 first so readers can get a sense of the absolute differences between the various approaches and then dive in to the description of various errors.

Reference: Jarvis, P.G. and McNaughton, K.G., 1986. Stomatal control of transpiration: scaling up from leaf to region. In Advances in ecological research (Vol. 15, pp. 1-49). Academic Press.

---

## Author Response (AR1)

**Point-by-Point Responses to Referee Comments**

Response by the authors (Wehr and Saleska, hereafter WS) to Anonymous Referee #1 (hereafter AR1):

AR1: Overall, this is an innovative study and should be considered for publication in the Biogeosciences after some revision. I have a few general comments and suggestions.

WS: Thank you for taking the time to review our manuscript, and for the helpful suggestions, to which we respond below.

AR1: The readability of this paper could be improved. I found that Methods, Results, and Discussion are difficult to follow. I had to go back and forth several times to find the necessary details and information. I understand this type of paper might not necessarily follow the same structure of a typical research paper, but I suggest the authors should clearly ad structurally lay out the data used and steps taken upfront. Potentially, an overview paragraph summarizing the study design, a table listing the different simulation scenarios, and/or a more explicit subtitle might also help readers. Figure legends, especially Figure 1 & 2, should be more self-explanatory.

WS: We apologize for the lack of clarity in the original manuscript. We have prepared a revised version that includes substantial changes aimed at making the presentation more straightforward and clear, including: (i) revised and additional text in all sections, (ii) a brief summary of the study design at the end of the introduction, (iii) reorganized Methods and Results sections with separate subsections for the simulations and the real time series analysis, and (iv) simplified figures. We believe the new version is much easier to follow.

AR1: Several recent studies suggested that the energy imbalance issue was likely caused by mesoscale or secondary circulations instead of instrumental or other local sources, and H and LE might be influenced disproportionally (Mauder et al., 2020; Xu et al., 2020). It might still be an open question, but I suggest the authors taking that into consideration.

WS: This is a very good point. Whether the Bowen ratio is preserved by EC measurement error is an open question. Our new manuscript version includes new text in the introduction, methods, and results sections dealing with this point, and backs away from recommending a specific flux correction method. Instead, the new manuscript focuses on comparison between the flux-gradient and inverted Penman-Monteith equations, and explores the impact of ideal and flawed corrections. It thereby motivates future work on the question of how to best correct the eddy fluxes.

AR1: It's a bit puzzling to me about one of the key arguments – the preferred use of flux gradient equations instead of Penman-Monteith equation. I think ultimately the main difference resulted from how the energy imbalance was treated and/or how the total available energy was partitioned. The psychrometric approximation should have only marginal influence, right? Or, do the authors imply anything additionally? For example, some studies used available energy

**(LE+H) or adjusted total energy in the Penman-Monteith equation. Would it be sufficient enough?**

WS: Our new version with revised introductory context, framing, and figures should help clarify this issue, which is central to the paper. The main practical difference between the FG and inverted PM equations is that the former uses measurements of H and LE while the latter uses measurements of A and LE, and then implicitly infers H as the difference between the two. Thus the FG equations involve underestimated H and LE, which bias stomatal conductance in opposite directions, whereas the inverted PM equation involves underestimated LE and overestimated H, which both bias stomatal conductance in the same direction. The psychrometric approximation is less important but not negligible (bias ~ 5%).

When people use the EC-derived available energy (H + LE) in the "Penman-Monteith equation", that is not the Penman-Monteith equation anymore; it is an undoing of the Penman-Monteith equation that moves back towards the FG equations on which it was based (which we are recommending use of here). The distinguishing feature of the PM equation is its elimination of H and surface (i.e. leaf) temperature.

All these points are explained more clearly in our new manuscript version. For example, our new introduction includes:

"The original (not inverted) Penman-Monteith equation was designed to estimate transpiration from the available energy (A), the vapor pressure deficit, and the stomatal and aerodynamic conductances. It was derived from simple flux-gradient relationships for LE and for the sensible heat flux (H) but was formulated in terms of A and LE rather than H and LE."

And our new Results and Discussion section includes:

"...the FG formulation (solid black lines) is always more accurate than the iPM formulation (solid red lines) because regardless of whether the gap is due to negative measurement bias in A or in H + LE, the iPM equation implicitly overestimates H (as the residual of the other fluxes) and therefore the leaf temperature and therefore the water vapor gradient, which exacerbates underestimation of the conductance. In other words, it is better to have both LE and H underestimated (as in the FG equations) than to have LE underestimated and H overestimated (as in the iPM equation)."

**Response by the authors (Wehr and Saleska, hereafter WS) to Anonymous Referee #2 (hereafter AR2):**

AR2: Wehr and Saleska identify the implication of residuals in the energy balance at flux sites and its implications on the calculations of stomatal conductance. The authors are well posed to tackle this problem given their previous work on stomatal conductance modelling. While the issue is important, they falter in the motivation of the study and ignore very carefully laid out theory regarding movement of water between leaves and the atmosphere.

WS: Thank you for reviewing our manuscript. We appreciate the perspective you have brought, which we believe to be complementary rather than contradictory, as we detail below.

AR2: The authors state that the Penman-Monteith equation was developed because sensible heat fluxes (H) were immeasurable at the time (while those of latent heat were not). This is not true, since both H and LE have previously been inferred from flux-gradient approaches or Bowen-ratio based approaches, and were prone to similar errors.

WS: To be fair, we said "difficult to measure" (in contrast to eddy flux sites, where H is a routine measurement), not "immeasurable". But we see how this sentence can be misleading, and so we will rephrase to say: "The original (not inverted) Penman-Monteith equation was designed to estimate transpiration from the available energy (A), the vapor pressure deficit, and the stomatal and aerodynamic conductances. It was derived from simple flux-gradient relationships for LE and for the sensible heat flux (H) but was formulated in terms of A and LE rather than H and LE."

AR2: In the abstract the authors note ". . .even though H is measured at least as accurately as LE at every EC site while the rest of the energy budget almost never is". This is also needs to be rephrased. Components of net radiation (Rnet) are routinely measured, and in fact, this is generally a more reliable measurement than the eddy flux of latent and sensible heat, which is prone to well-known errors (e.g. poor turbulence).

WS: We agree (and the manuscript states explicitly) that Rnet is ubiquitously measured. We meant this sentence to refer to the rest of the energy budget taken all together, including storage and ground heat flux. That entire "rest of the energy budget" is almost never measured. We will rephrase to say: "EC sites, in contrast, measure H and LE but rarely assess A in its entirety. True A is net radiation (Rn) minus heat flux to the deep soil (G), minus heat storage (S) in the shallow soil, canopy air, and biomass. …"

AR2: However, putting the issue of instrumentation aside, the differences between H + LE + G and Rnet is often due to differences in "fetch" or missing large eddies in tall forest canopies. 4 channel net radiometers are placed adjacent to the towers, and can be influenced by the tower itself, whereas measurements from IRGAs are highly contingent on the footprint. The authors note correctly that G (soil heat fluxes) are not universally measured, and areas sampled may not be representative of the average soil heat flux from the site.

WS: Yes, we agree that the footprint mismatch between radiometers and the eddy fluxes is another reason why relying on energy balance (i.e. on the inverted PM equation) to estimate stomatal conductance is problematic at eddy flux sites, and we will mention it in our revised manuscript. We also agree that much evidence points to EC missing large, slow circulations, and our revised text will highlight this fact, e.g. by adding: "The other major contributor, which also impacts the iPM equation, is systematic underestimation of H + LE by the EC method, probably due to its failure to capture sub-mesoscale transport (Foken, 2008; Stoy et al., 2013; Charuchittipan et al., 2014; Gatzsche et al., 2018; Mauder et al., 2020)." Our revised manuscript will focus on the difference between the FG and inverted PM results without depending on a particular explanation for the energy budget gap. AR2: Coming to the main argument of the paper, the Penman-Monteith equation accounts for the fact that water loss from landscapes is controlled by biotic and abiotic factors. Leaves can reduce the width of stomatal apertures to limit water loss. However, the total evaporation from terrestrial ecosystems, especially leaves, is also a function of available energy (Rnet). This framework was developed in a seminal paper by Paul Jarvis and Keith McNaughton (1986) where they proposed a decoupling coefficient that determines the extent to which transpiration is "stomatally imposed". By using a simple flux-gradient theory the authors imply that transpiration is totally stomatally imposed. This argument is likely to work in tall rough canopies where exchange of momentum (and therefore scalars) with the well-mixed air above is efficient. However, at sites where roughness is low and canopies are homogenous, this approach is more likely to lead to erroneous estimates of stomatal conductance, since it is the available energy that will dominate the amount of water vapour that is lost to the atmosphere. Of course, to an extent, this problem is mitigated by also inferring boundary layer conductance and sharing the limitation of water loss between stomata and resistance imposed by the boundary layer.

WS: Thank you for the reminder of the important paper by Jarvis and McNaughton (hereafter JM86), which we will now use to better contextualize our revised manuscript, and which we hope makes clear that we are not, in fact, implying "that transpiration is totally stomatally imposed." The crux of Jarvis and McNaughton 1986 (hereafter JM86) is that stomata are only one segment of the transpiration pathway between the substomatal cavity and the atmosphere (a point that goes back at least to Monteith 1965). The idea of JM86 can be concisely summed up in the flux-gradient framework by saying that the total resistance to transpiration consists of the stomatal resistance (r s) plus an aerodynamic resistance (r a) that includes all resistance between the surface of the leaf and the chosen atmospheric reference point, and which therefore increases relative to r s as the spatial scale increases (because as spatial scale increases, the reference point moves from outside the leaf boundary layer to outside the canopy to outside the planetary boundary layer, progressively incorporating more aerodynamic resistance into r a). The simple flux-gradient equation, transpiration = gradient/(r + r = a), tells us straightaway that the FG makes no particular *a priori* assumption about the relative importance of stomatal resistance: if r s dominates over r a, then transpiration will be sensitive to r s, whereas if r a dominates over r s, then transpiration will be insensitive to r s (in which case it is said to be limited by available energy, which is really just another way of saying it is limited by r a rather than by r s). That is the conclusion of JM86, paraphrased. The only assumption about the degree of stomatal limitation creeps into our analysis when we consider conditions under which the leaf boundary layer resistance takes on a typical forest value (with low decoupling coefficient) and neglect the rest of the aerodynamic resistance (according to the discussion on lines 138-143).

For our revised manuscript, we will discuss the issue of non-stomatal limitation and include simulations we have done showing that if the aerodynamic resistance increases (i.e. the decoupling coefficient increases), then the FG equations actually become more accurate while the inverted PM equation becomes less accurate. That is because decoupling (i.e. large aerodynamic resistance) impedes the exchange of heat and so causes the leaf temperature to increase, which causes the saturation vapor pressure inside the leaf to increase even faster (nonlinearly, according to the Clausius-Clapeyron relation). Thus transpiration actually increases and the Bowen ratio approaches 0 (so that underestimation of H becomes unimportant). The

psychrometric approximation also becomes poorer in this situation because it is a linearization of the Clausius-Clapeyron relation, adding additional bias to the inverted PM equation. This "calm limit" and its misrepresentation by the PM equation is thoroughly discussed in a new paper that just came out (and which we now also cite): McColl, K. A. (2020). Practical and theoretical benefits of an alternative to the Penman-Monteith evapotranspiration equation. Water Resources Research, 56, e2020WR027106. https://doi.org/10.1029/2020WR027106. Of course both the FG and inverted PM equations rely on an estimate of the aerodynamic resistance, and they become increasingly sensitive to it as it becomes increasingly limiting.

AR2: Penman-Monteith never intended to solve for stomatal conductance, rather to estimate water loss from vegetated canopies in a way that eliminated the need to know surface variables e.g. surface temperature. The method includes the parametrization of a "surface conductance" which really is somewhat of an emergent property since its accounts for a cumulative effect of "all stomata" but also canopy structure and coupling (much like a canopy scale stomatal conductance or gsv in the current manuscript).

WS: Indeed, the PM equation was developed to estimate transpiration, not to solve for stomatal conductance. Nonetheless, use of the inverted PM equation to calculate stomatal conductance from transpiration is common in the scientific literature and goes back at least 25 years (to Grace et al., Glob. Change Biol., 1995). Our purpose in this paper is to address that common literature practice, and to show that if you are going to calculate stomatal conductance from transpiration, then you are better off using the flux-gradient (FG) equations than the inverted PM equation. The fundamental difficulty you raise applies similarly regardless of which equations are used: when transpiration is insensitive to stomatal conductance (i.e. when the decoupling coefficient is close to 1), there will naturally be very large error in the stomatal conductance you retrieve from transpiration. In our revised introduction, we will more clearly frame the scientific task under consideration, including this fundamental limitation, and we will distinguish the original PM equation and its purpose from the inverted PM equation commonly used to estimate stomatal conductance from transpiration.

Our revised first paragraph will begin: "Leaf stomata are a key coupling between the terrestrial carbon and water cycles. They are a gateway for carbon dioxide and transpired water and often limit both at the ecosystem scale (Jarvis and McNaughton, 1986)."

And our revised second paragraph will begin: "When the aerodynamic conductance to water vapor outside the leaf (gaV) is greater than gsV, the latter exerts a strong influence on transpiration, from which it can be inferred. The standard method is to calculate gsV from eddy covariance (EC) measurements of the latent heat flux (LE) via the inverted Penman-Monteith (iPM) equation (Monteith, 1965; Grace et al., 1995) — but the EC method and the iPM equation make a strange pairing. The original (not inverted) Penman-Monteith equation was designed to estimate transpiration from the available energy (A), the vapor pressure deficit, and the stomatal and aerodynamic conductances. It was derived from simple flux-gradient relationships for LE and for the sensible heat flux (H) but was formulated in terms of A and LE rather than H and LE."

AR2: I do really appreciate the careful analyses that the authors have performed and a nice example of this is highlighting the errors in the psychometric approximation, when FG and PM methods are considered equivalent.

WS: Thank you; we're glad that point was of interest. You might be interested in the new paper we mentioned above (McColl, 2020), which is all about how the psychrometric approximation leads to significant error and incorrect limiting behavior in the (not inverted) PM equation.

AR2: Thus, in my opinion, the authors should need to reconcile errors due to flux-gradient approaches with Jarvis and McNoughton (1986). I would then consider this a very significant contribution to the literature and fit to be published in Biogeosciences.

WS: Hopefully our responses above have convinced you that the flux-gradient framework is entirely consistent with JM86, and that it does not invoke any additional error compared to the inverted PM equation.

AR2: A minor note on figures: I think Figures 1 and 2 are a little complicated and could be simplified. It might help to even show figure 3 first so readers can get a sense of the absolute differences between the various approaches and then dive in to the description of various errors.

WS: We have created new, simplified versions of our figures, in which only two flux correction scenarios are included. We have also created a small figure that aids with interpretation of Fig. 1 by visually highlighting the contribution of each source of bias to the inverted PM equation.

**Calculating Canopy Stomatal Conductance from Eddy Covariance Measurements, in Light of the Energy Budget Closure Problem**

Richard Wehr1, Scott. R. Saleska1

1Ecology and Evolutionary Biology, University of Arizona, Tucson, 85721, U.S.A.

5 Correspondence to: Richard Wehr (rawehr@email.arizona.edu)

Abstract. Canopy stomatal conductance is commonly estimated from eddy covariance measurements of the latent heat flux (*LE*) by inverting the Penman-Monteith equation. That method ignores eddy covariance measurements of the sensible heat flux (*H*) and instead calculates  $\underline{H}$  implicitly as the residual of all other terms in the site energy budget. Here we show that canopy stomatal conductance is more accurately calculated from EC measurements of both *H* and *LE* using the flux-gradient

10 equations that define conductance and underlie the Penman-Monteith equation, especially when the site energy budget fails to close due to pervasive biases in the eddy fluxes and/or the available energy. The flux-gradient formulation dispenses with unnecessary assumptions, is conceptually simpler, and is as or more accurate in all plausible scenarios. The inverted Penman-Monteith equation, on the other hand, contributes substantial biases and erroneous spatial and temporal patterns to canopy stomatal conductance, skewing its relationships with drivers such as light and vapor pressure deficit.

**15 1 Introduction**

Leaf stomata are a key coupling between the terrestrial carbon and water cycles. They are a gateway for carbon dioxide and transpired water and often limit both at the ecosystem scale (Jarvis and McNaughton, 1986). Although the many stomata in a plant canopy experience a wide range of micro-environmental conditions and therefore exhibit a wide range of behaviors at any given moment in time, it has proven useful in many contexts to approximate the canopy as a single 'big leaf' with a single

- 20 stoma (Baldocchi et al., 1991; Wohlfahrt et al., 2009; Wehr et al., 2017). That stoma is characterized by the canopy stomatal conductance to water vapor  $(g_{st})$ , which can be defined as the total canopy transpiration divided by the transpiration-weighted average water vapor gradient across the many real stomata. This canopy stomatal conductance is not a simple sum of the individual leaf-level conductances and does not vary with time or environment in quite the same way as they do (Baldocchi et al., 1991); it is impacted, for example, by changes in the distribution of light within the canopy.
- 25

.

When the aerodynamic conductance to water vapor outside the leaf  $(g_{aF})$  is greater than  $g_{aF}$ , the latter exerts a strong influence on transpiration, from which it can be inferred. The standard method is to calculate  $g_{aF}$  from eddy covariance (EC) measurements of the latent heat flux (*LE*) via the inverted Penman-Monteith (iPM) equation (Monteith, 1965; Grace et al., 1995) — but the EC method and the iPM equation make a strange pairing. The original (not inverted) Penman-Monteith

1

**Deleted: (EC) Deleted: h (PM) Deleted: implicitly represents Formatted: Font: Italic **Deleted:** — even though H is measured at least as accurately as LE at every EC site while the rest of the energy budget almost never is. We argue Deleted: gsV Deleted: should Deleted: instead be Deleted: Deleted: formulation Deleted: s Deleted: s Deleted: PM Deleted: n **Deleted:** provides more accurate values of $g_{sV}$ for Deleted: in which the measured energy budget fails to close, as is common at EC sites Deleted: PM Deleted: g.v Deleted: To minimize the impact of the energy budget closure problem on the PM equation, it was previously proposed that the eddy fluxes should be corrected to close the long-term energy budget**

problem on the PM equation, it was previously proposed that the eddy fluxes should be corrected to close the long-term energy budget while preserving the Bowen ratio (B = HL/LE). We show that such a flux correction does not fully remedy the PM equation but should produce accurate values of  $g_{st}$  when combined with the flux-gradient formulation.

**Deleted: the main**

| Deleted: , simultaneously controlling |
|---------------------------------------|
| Deleted: the                          |
| Deleted: passage                      |
| Deleted: of                           |
| Deleted: T                            |
| Deleted: ; yet                        |
| Deleted: e                            |
| Deleted: aggregate                    |
| Deleted: is then                      |
| Deleted: ¶                            |
| Deleted: estimate                     |
| Deleted: is                           |

equation was designed to estimate transpiration from the available energy (*A*), the vapor pressure deficit, and the stomatal and aerodynamic conductances. It was derived from simple flux-gradient relationships for *LE* and for the sensible heat flux (*H*) but was formulated in terms of *A* and *LE* rather than *H* and *LE*. Thus the inverted PM equation estimates  $g_{SV}$  from *A* and *LE* rather than from *H* and *LE*. EC sites, in contrast, measure *H* and *LE* but rarely assess *A* in its entirety. True *A* is net radiation

- 75 (*Rn*) minus heat flux to the deep soil (*G*), minus heat storage (*S*) in the shallow soil, canopy air, and biomass. In wetland ecosystems, heat flux by groundwater discharge (*W*) can also be important (Reed et al., 2018). While net radiation measurements are ubiquitous at EC sites, ground heat flux measurements are less common (Stoy et al., 2013; Purdy et al., 2016) and heat storage and discharge measurements are rare (Lindroth et al., 2012; Reed et al., 2018). As such it is common practice to simply omit *S* and *W* and sometimes *G* from *A* in the iPM equation.
- 80

In general, neither *S* nor *G* is negligible. Insufficient measurement of *S* in particular has been shown (Lindroth et al., 2010; Leuning et al., 2012) to be a major contributor to the infamous energy budget closure problem at EC sites, which is that the measured turbulent heat flux H + LE is about 20% less than the measured available energy  $R_n - G$  on average across the FLUXNET EC site network (Wilson et al., 2002; Foken, 2008; Franssen et al., 2010; Leuning et al., 2012; Stoy et al., 2013).

- 85 The other major contributor, which also impacts the iPM equation, is systematic underestimation of H + LE by the EC method, probably due to its failure to capture sub-mesoscale transport (Foken, 2008; Stoy et al., 2013; Charuchittipan et al., 2014; Gatzsche et al., 2018; Mauder et al., 2020), Leuning et al. (2012) assessed the relative contributions of *S* and H + LE to the closure problem using the fact that *S* largely averages out over 24 hours while  $R_n$ , H, and LE do not; thus *S* contributes to the hourly but not the daily energy budget (Lindroth et al., 2010; Leuning et al., 2012). Analyzing over 400 site-years of data, they
- 90 found that the median slope of *H* + *LE* versus *Rn G* was only 0.75 when plotting hourly averages but went up to 0.9 when plotting daily averages. This result suggests that for the average FLUXNET site, 60% of the energy budget gap is attributable to *S* and 40% to *H* + *LE*. Depending on the depth at which *G* is measured (which is not standard), *G* might also average down considerably over 24 hours and thereby share some of the 60% attributed to *S*. Conversely, the part of *G* that does not average out over 24 hours might share some of the 40% attributed to *H* + *LE*, as might *Rn* and *W*. Part of that 40% might also be due 95 to mismatch between the view of the net radiometer and the flux footprint of the eddy covariance tower. But *S* and *H* + *LE* are the most likely sources of large systematic bias across sites.

The iPM equation is further impacted by how the underestimation of H + LE is partitioned between H and LE. While some studies have reported that underestimation of H + LE roughly preserves the Bowen ratio ( $B_{a} = H(LE)_{a}$ , others have reported that 100 the failure to capture sub-mesoscale transport causes EC to underestimate H more than LE (Mauder et al., 2020) — a situation that would benefit the iPM equation. Charuchittipan et al. (2014) quantified the preferential underestimation of H relative to LE using a simple formula based on the buoyancy flux, and a study of tall vegetation suggested that the formula holds when B is high (B > 2) but that B is instead preserved when it is low or moderate (B < 1.5) (Gatzsche et al., 2018). In the latest review

**Deleted: on average across Deleted: error Formatted: Font: Not Italic Formatted: Font: Not Italic Formatted: Font: Not Italic Formatted: Font: Not Italic Formatted: Font: Not Italic**

of the issue, Mauder et al. (2020) concluded that recent evidence "tends towards a partitioning somewhere between a buoyancyflux-based and a Bowen-ratio-preserving" one.

110

To deal with the energy budget closure problem, Wohlfahrt et al. (2009) considered various schemes for correcting the fluxes in the iPM equation, following earlier recommendations that EC fluxes be corrected to close the energy budget in a more general context (Twine et al., 2000). All but one of the schemes in Wohlfahrt et al. (2009) involve attributing the half-hourly budget gap entirely to A or entirely to H + LE, neither of which is generally realistic according to the subsequent results of

115 Leuning et al. (2012), mentioned above. The remaining option from Wohlfahrt et al. (2009) increases *H* and *LE* to close the long-term (e.g. daily or monthly) budget gap while preserving the Bowen ratio (*B*), which is in line with Leuning et al (2012) in that it attributes the long-term gap to EC and the remaining gap to storage.

Here we use data simulations to show that regardless of whether the energy budget gap is due to A or H + LE, and regardless of how the EC bias is partitioned between the buoyancy-flux and Bowen-ratio limits, stomatal conductance is more accurately

- obtained by direct application of the two simple flux-gradient (FG) equations on which the iPM equation is based than by use of the iPM equation itself. By using simulations, we can know the "true" target values and hence the absolute biases in gase. We also use our simulations to test the effects of perfect and imperfect eddy flux corrections, and of bias in the aerodynamic conductance outside the leaf. Lastly, we leave the simulations behind and show how the discrepancy between the FG and iPM
- 125 formulations impacts the retrieval of gs, over time using real measurements from a conifer forest. We present the FG and iPM formulations in Section 2, describe our methods for comparing them in Section 3, and report our findings in Section 4.

By definition, conductance is the proportionality coefficient between a flux and its driving gradient. In the case of  $g_{SV}$ , the flux

is transpiration and the gradient is the vapor pressure differential across the "big-leaf" stoma. It is therefore relatively

**2 Theory**

is transplation and the gradient is the vapor pressure differential across the organized solution solution is transplation and sensible heat (Baldocchi et al., 130 straightforward to calculate  $g_{S'}$  from the flux-gradient (FG) equations for transpiration and sensible heat (Baldocchi et al.,

1991), rearranged as follows (Wehr and Saleska, 2015):

- $r_{sV} = \frac{e_s(T_L) e_a}{RT_a E} r_{aV}$
- 135  $T_L = \frac{Hr_{aH}}{\rho_a c_p} + T_a$

where  $r_{sV}$  (s m-1) is the stomatal resistance to water vapor,  $r_{sV}$  is the aerodynamic resistance to water vapor (s m-1),  $r_{sV}$  is the aerodynamic resistance to heat (s m-1), *E* is the flux of transpired water vapor (mol m-2 s-1), *H* is the sensible heat flux (W m-1)

**Deleted:**

**Deleted: The advantage of**

**Deleted: are known**

**Deleted: 1**

On account of the energy budget closure problem, it was proposed a decade ago that the PM equation should use values of H and LE that have been adjusted (explicitly or implicitly) to close the energy budget while preserving the Bowen ratio B = H/LE (Wohlfahrt et al., 2009). Preservation of the Bowen ratio is advisable because the most likely causes of pervasive underestimation by EC (such as anenometer tilt, advective loss, and large-scale air motions) afflict H and LE proportionally or nearly so. On the other hand, attributing the entire budget gato H + LE is inadvisable given the above evidence for the importance of  $S^6$

Rather than trying to kluge an accurate method for using the PM equation at EC sites, we propose that the underlying flux-gradient (FG) equations for sensible heat and water vapor should be applied directly to the calculation of  $g_{s'}$  (as in Baldocchi et al., 1991; Wehr and Saleska, 2015). As we will show, the FG equations take advantage of the H measurements provided by EC, do not require measurements of  $R_{a}$ , G, S, or W, and are universally more accurate than the PM equation by a substantial margin.4

**Deleted: equations**

(1)

(2)

| Deleted: at                  |  |
|------------------------------|--|
| Deleted: b                   |  |
| Deleted: b                   |  |
| Deleted: b                   |  |
| Deleted: leaf boundary layer |  |
| Deleted: b                   |  |
| Deleted: leaf boundary layer |  |

2),  $T_a$  is the air temperature (K),  $T_L$  is the effective canopy-integrated leaf temperature (K),  $\rho_a$  is the density of (wet) air (kg m-3),  $c_p$  is the specific heat capacity of (wet) air (J kg-1 K-1),  $e_a$  is the vapor pressure in the air (Pa),  $e_3(T_L)$  is the saturation vapor

175 pressure inside the leaf as a function of  $T_L$  (Pa), and R is the molar gas constant (8.314472 J mol-1 K-1). The equation for the saturation vapor pressure (Pa) as a function of temperature (K) is (World Meteorological Organization, 2008):

$$e_{s}(T) = 611.2e^{\left(\frac{17.62(T-273.15)}{243.12+(T-273.15)}\right)}$$

(3)

(4)

(5)

180 The aerodynamic resistances describe the path between the surface of the "big leaf" and whatever reference point in the air at which *Ta*, *pa*, and *cp* are measured. If that reference point is the top of an eddy flux tower, then that path includes the leaf boundary layer (through which transport is quasi-diffusive) as well as the canopy airspace and some above-canopy air (through which transport is turbulent). The turbulent eddy resistance (*re*) anay be calculated by various methods that do not agree particularly well with one another (e.g. see Baldocchi et al., 1991; Grace et al., 1995; Wehr and Saleska, 2015), but is typically small in "rough surface" ecosystems like forests during the daytime, when *raft* and *raf* tend to be dominated by the leaf boundary layer resistances *rbiff* and *rbiff* An empirical model such as the one given in the Appendix can be used to calculate *rbiff* as a function of wind speed and other variables. Using that model in a temperate deciduous forest, *rbiff* was found to vary only between 8 and 12 s m-1 (Wehr and Saleska, 2015), and so we simply take it to be constant at 10 s m-1 here. The corresponding resistance to water vapor transport can be calculated from *rbiff* via (Hicks et al., 1987):

190

195

$$r_{bV} = \frac{1}{f} r_{bH} \left(\frac{Sc}{Pr}\right)^{\frac{2}{3}}$$

where *Sc* is the Schmidt number for water vapor (0.67), *Pr* is the Prandtl number for air (0.71), and *f* is the fraction of the leaf surface area that contains stomata (f = 0.5 for hypostomatous leaves, which have stomata on only one side, and f = 1 for amphistomatous leaves, which have stomata on both sides). The aerodynamic resistances to sensible heat and water vapor are then  $r_{all} = r_{bll} + r_{sc}$  and  $r_{all} = r_{bl} + r_{sc}$ .

Finally, the stomatal conductance to water vapor (mol m-2 s-1) is obtained from  $r_{sV}$  by (Grace et al., 1995):

$$\quad g_{sv} = \left(\frac{P}{RT_L}\right) \frac{1}{r_{sv}}$$

where P is the atmospheric pressure.

| Deleted: canopy |  |
|-----------------|--|
| Deleted: the    |  |
| Deleted: canopy |  |
| Deleted: the    |  |
| Deleted: canopy |  |
| Deleted: canopy |  |

| Formatted: | Font: | Italic, | Subscript |
|------------|-------|---------|-----------|
|------------|-------|---------|-----------|

**Moved (insertion) [1]**

**Deleted:** Also note that in the above equations,  $T_a$ ,  $e_a$ ,  $\rho_a$ , and  $c_p$  refer to the canopy air; that is, the air just outside the leaf boundary layer. In practice, these variables may be measured above the canopy and converted to canopy air values via the turbulent eddy or aerodynamic transport resistance, which

**Deleted:**

| Deleted: ¶   |  |
|--------------|--|
| Deleted: 1   |  |
| Deleted: The |  |

The above FG theory is also the basis of the Penman-Monteith equation for a leaf (Monteith, 1965), and its inverted form, (Grace et al., 1995), which can be expressed as:

225
$$r_{sV} = \frac{s(R_n - G - S - W - LE_w)r_{aH} + \rho_a c_p(e_s(T_a) - e_a)}{\gamma LE_w} - r_{aV}$$

where  $LE_{\rm tr}$  is the latent heat flux associated with transpiration (W m-2),  $LE_{\rm ev}$  is the latent heat flux associated with evaporation that does not pass through the stomata (W m-2),  $e_s(T_a)$  is the saturation vapor pressure of the air as a function of  $T_a$  (Pa) rather than  $T_{L}$ , s is the slope of the  $e_s$  curve at  $T_a$  (Pa K-1), and  $\gamma$  is the psychrometric constant at  $T_a$  (Pa K-1).  $R_n$ , G, S, and W also have units of W m-2. Latent heat flux is water vapor flux (E) times the latent heat of vaporization of water (about 44.1 × 103 J mol-1).

The inverted PM equation is usually expressed in a slightly simpler form by neglecting the distinctions (a) between transpiration and evaporation, and (b) between the leaf boundary layer resistances to heat and water vapor. We retain those

- 235 distinctions here in order to highlight two important points:
  - Absent a means to accurately partition the measured eddy flux of water vapor into transpiration and non-stomatal evaporation (e.g. from soil or wet leaves), the FG and iPM equations are applicable only when evaporation is negligible, which is a difficult situation to verify but does occur at particular times in particular ecosystems (see, e.g., Wehr et al., 2017).
  - 2. Setting  $r_{bV} = r_{bH}$  instead of using Eq. (4) is a good approximation for amphistomatous leaves (stomata on both sides) but a poor approximation for the more common hypostomatous leaves (stomata on only one side) (Schymanski and Or, 2017). Indeed, we find that if  $r_{bV}$  is set equal to  $r_{bH}$  for hypostomatous leaves, the iPM equation underestimates  $g_{sV}$  by about 10% (depending on the relative resistances of the stomata and boundary layer) even when the site energy budget is closed.

Note that the iPM equation can be derived from the FG equations by invoking energy balance to replace H with A - LE in Eq. (2) and then linearizing the Clausius-Clapeyron relation to eliminate leaf temperature;

 $250 \quad s \approx \frac{e_s(T_a) - e_s(T_L)}{\sqrt{T_a - T_L}} \Rightarrow e_s(T_L) \approx e_s(T_a) - s(T_a - T_L) = e_s(T_a) + s\left(\frac{(A - LE)T_{aH}}{\rho_a c_p}\right)$

240

245

This psychrometric approximation has been shown to cause significant bias and incorrect limiting behavior in the Penman-Monteith equation (McColl, 2020). McColl (2020) derived a similar, alternative equation that remedies those problems but still uses measurements of A instead of H. The psychrometric approximation and the substitution for H are the only two

| Deleted: The                                  |  |
|-----------------------------------------------|--|
| Deleted: (PM)                                 |  |
| Deleted: , w                                  |  |
| Deleted: hen                                  |  |
| Deleted: to solve for the stomatal resistance |  |

(6)

(7)

| 1                 | Deleted: eliminate                              |
|-------------------|-------------------------------------------------|
| /)                | Formatted: Font: Italic                         |
| A                 | Deleted: from                                   |
|                   | Formatted: Font: Italic                         |
|                   | Deleted: by                                     |
|                   | Deleted: making the psychrometric approximation |
| Λ                 | Deleted: T                                      |
|                   | Deleted: T                                      |
|                   | Deleted: T                                      |
| $\mathbb{C}^{+}$  | Deleted: T                                      |
| $\langle \rangle$ | Deleted: $\rightarrow \rightarrow \rightarrow$  |
| N                 | Deleted: ¶                                      |
|                   | Deleted: ose                                    |

270 differences between the FG and iPM formulations. Both formulations rely on the same water flux measurements to estimate transpiration, both approximate the canopy as a 'big leaf', and both use the same estimate of aerodynamic resistance.

**3 Methods**

Our analysis consisted of two parts: simulations and real data analysis. The simulations were designed to unambiguously demonstrate the impact of flux measurement biases and the resultant energy budget gap on FG and iPM calculations of  $g_{sV}$ , as

275 well as to test the sensitivity of gsF to bias in the estimated aerodynamic resistance outside the leaf; they are described in Section 3.1. The real data analysis was designed to assess the magnitude and temporal variation of the discrepancy between the FG and iPM formulations in a real forest and is described in Section 3.2.

**3.1 Simulations**

- We assessed the proportional bias in gs/ calculated via the iPM and FG formulations by simulating observations and using them to estimate gs/. The simulations were of three snapshots in time roughly typical of midday in three different ecosystems: a temperate deciduous forest in July (the Harvard Forest in Massachusetts, USA; Wehr et al., 2017), a tropical rainforest in May (the Reserva Jaru in Rondônia, Brazil; Grace et al., 1995), and a tropical savannah in September (Virginia Park in Queensland, Australia; Leuning et al., 2005). The purpose of including three different ecosystems was to test the FG and iPM formulations across a broad range of environmental and biological input variables (especially Bowen ratios), not to provide a
- 285 lookup table of quantitative  $g_{SV}$  corrections for other sites. The particular sites and time periods within each ecosystem were chosen merely for convenience, as the requisite variables were readily obtainable from the literature or from our past work.

Next, we simulated a wide range of measurement bias scenarios, each with a 20% gap in the energy budget (the FLUXNET
 average). The simulations were explored along three main axes of variation;

**Deleted: and**

**Deleted:**

**Moved up [1]:** Also note that in the above equations,  $T_a$ ,  $e_a$ ,  $\rho_a$ , and  $e_p$  refer to the canopy air; that is, the air just outside the leaf boundary layer. In practice, these variables may be measured above the canopy and converted to canopy air values via the turbulent eddy or acrodynamic transport resistance, which may be calculated by various methods that do not agree particularly well with one another (e.g. see Baldocchi et al., 1991; Grace et al., 1995; Wehr and Saleska, 2015). Fortunately, that resistance is often much smaller than the stomatal and leaf boundary layer resistances during the day and can sometimes be neglected in the calculation of  $g_{ij'}$ .

**Deleted: unequivocally**

**elecent percer**

[revised manuscript text omitted]

through late afternoon), it is almost certain that the aerodynamic resistance was much less than the stomatal resistance and therefore that the FG and iPM equations were insensitive to  $r_{bH}$  and  $r_{e}$  (see Section 4.1).

| 4 Results and Discussion | lts and Discussion |
|--------------------------|--------------------|
|--------------------------|--------------------|

**4.1 Absolute Biases Revealed by Simulations**

[revised manuscript text omitted]

without ...